# Selection of Short-Day Strawberry Genotypes through Multivariate Analysis

**DOI:** 10.3390/plants12142650

**Published:** 2023-07-14

**Authors:** Thiago Rutz, Juliano Tadeu Vilela de Resende, Keny Henrique Mariguele, Ricardo Antônio Zeist, Andre Luiz Biscaia Ribeiro da Silva

**Affiliations:** 1Department of Horticulture, Auburn University, Auburn, AL 36849, USA; trd0017@auburn.edu (T.R.); azb0207@auburn.edu (A.L.B.R.d.S.); 2Departament of Agronomy, Universidade Estadual de Londrina, UEL, Rodovia Celso Garcia, km 380, Londrina 86051-900, PR, Brazil; 3Experimental Station, Empresa de Pesquisa Agropecuária e Extensão Rural de Santa Catarina, Epagri, Rodovia Antônio Heil, 6800, Itajaí 88318-112, SC, Brazil; kmariguele@hotmail.com; 4Department of Agronomy, Universidade Estadual do Centro Oeste, Unicentro, Alameda Élio Antonio Dalla Vecchia, 838, Guarapuava 85040-167, PR, Brazil; ricardo-zeist@bol.com.br

**Keywords:** *Fragaria × ananassa*, genetic parameters, selection index, clustering

## Abstract

Strawberries are produced in tropical regions using imported cultivars adapted to temperate and subtropical climates. These cultivars, under tropical conditions, produce below their genetic potential. Through multivariate analyses, the objective was to evaluate and select short-day strawberry genotypes based on intraspecific crosses, product characteristics, and fruit quality. The genotypes were obtained from the cross between ‘Camino Real’ (female parent) and the first-generation genotypes RVCA16, RVCS44, RVFS06, RVFS07, and RVDA11 (male parent), obtained in previous selections. The experimental design consisted of augmented blocks with standard controls, consisting of first-generation genotypes and commercial cultivars. The fruits were harvested and evaluated for productivity and post-harvest characteristics: total fruit mass (MTF), total number of fruits (TFN), average fruit mass (AFM), commercial fruit mass (CFM), fruit commercial number (CFN), average commercial mass of fruits (ACFM), total soluble solids (TSS), firmness (F), brightness (L), hue angle (°Hue), and chroma (C). The selection index of Mulamba and Mock (1978) was used with an intensity of 3% to obtain superior genotypes and submitted to multivariate analysis for comparative purposes. Of the 1500 genotypes evaluated, it was possible to select 44 genotypes with characteristics superior to the 13 controls. The RVDA11CR59 genotype showed better values for the attributes of interest, but the RVCS44CR population, from the cross between ‘Camino Real’ × RVCS44 (‘Camarosa’ × ‘Sweet Charlie’), obtained the highest number (16) of individuals among those selected. Significant traits had high heritability but were not necessarily reflected in high selection gain. Coefficients of genetic variation were high, indicating sufficient genetic variability to select genotypes for these traits. When multivariate analyses were used, it was possible to group the selected genotypes into the same cluster according to the similarity and balance in the responses to the evaluated variables, demonstrating that these analyses help other parameters choose superior genotypes. The multivariate analysis allowed the selection of more balanced genotypes for production and post-harvest traits for tropical climates.

## 1. Introduction

The strawberry (*Fragaria* × *ananassa* Duch.) is one of the world’s most cultivated and consumed small fruits. Very tasty and aromatic, with an intense red color, it is rich in mineral nutrients and nutraceutical compounds, such as anthocyanins [1,2]. For fruiting, strawberry cultivars need interaction between photoperiod and temperature, classified as short-day, long-day, and day-neutral cultivars; the last group is influenced only by temperature.

In recent years, there has been a predominance of day-neutral cultivars, which theoretically produce year-round, providing the consumer with fresh fruit during spring and summer, but at a higher cost for both the consumer and the producer. With the advancement of these cultivars, breeding programs have invested less in developing short-day cultivars. However, in their shorter production period, these cultivars present productivity equivalent to day-neutral cultivars, which have a longer production cycle and demand higher phytosanitary and nutritional management costs. Some regions have opted for staggered production, with varieties that respond to a short and neutral photoperiod. The combined production of neutral and short-day cultivars provides greater profitability and productivity for the strawberry [3,4].

Short-day cultivars have high yield potential and larger fruits, but growers do not tend to produce them as day-neutral cultivars due to lower heat tolerance and shorter harvest periods [5,6]. Breeding programs in tropical regions have sought cultivars that can produce year-round [7,8,9,10] but without discarding the development of varieties responsive to a short photoperiod [11,12].

The search for short-day, productive cultivars with good post-harvest characteristics and adapted to tropical conditions is in focus in the strategic planning of breeding programs in countries with this climatic condition. However, in the early stages of the breeding program, identifying genotypes with good yield and physical–chemical quality (shape, color, brightness, firmness, flavor, and bioactive compounds) of the fruit is a complex and challenging process [6,13]. It should consider as many variables as possible and seek the best balance. Furthermore, estimating the genetic parameters involved in the crosses and the generated populations allows us to understand better which breeding and statistical methods can be more efficient for analyzing the generated data set.

The use of different methods that help the breeder in the selection and management of superior genotypes becomes essential. Thus, the use of multivariate methods can help in the selection efficiency. Mulamba and Mock’s (1978) [14] index showed promise in screening superior strawberry genotypes selected based on various attributes of economic importance [7,8,10]. Likewise, principal component analysis (PCA) has helped to verify the similarity between genotypes and commercial cultivars [7] for classification into groups according to their productive potential [15] and in the comparison of infrared spectra of strawberry juice regarding storage time [16]. Furthermore, when PCA was integrated with cluster analysis for physicochemical characterization, they proved to be good statistical tools for assessing compositional variations in fruits of different species [17].

Although multivariate analyses have properties such as integrating large amounts of data and are complementary to univariate statistical tests, such as the t-test or ANOVA, these methodologies can be complex to interpret for individuals unfamiliar with this type of analysis [18], given the scarcity of complementary studies between these different methodologies. Therefore, this study used multivariate analyses to select and evaluate short-day strawberry genotypes obtained from intravarietal crosses regarding the characteristics of interest (production and post-harvest) and genetic parameters.

## 2. Results

### 2.1. Genotypes Selection

Regarding the ANOVA results (Table 1) of the 11 traits evaluated, only the total number of fruits (TFN) and average fruit mass (AFM) were not significant (*p* ≤ 0.01), and, therefore, the genetic parameters were not calculated for these two attributes, just as they were not considered in other assessments. All the other nine characteristics obtained coefficient of variation values below 20%, indicating low variation for the average values.

Regarding the genetic parameters (Table 1), the values of the genotypic coefficient of variation (CVg) and the ratio between the genotypic and environmental variation coefficients (CVg/CVe) were high, indicating values ranging from 6.19% to 32.31% and between 1.08 and 11.84, respectively. Likewise, high heritability was observed for all characteristics, especially those related to fruit colors such as luminosity (L), hue angle (°Hue), and chroma (C), which presented the highest values for this parameter (99.29%, 91.03% and 93.70%, respectively).

Analyzing the estimates of absolute gains and selection percentages for the characters studied (Table 1), the averages between the selected populations (Xs) obtained higher values for most of the observed characters than the average of the controls (Xo). Except for C and TSS, these averages can be considered close. This can be observed for the evaluated genotypes (Table 1). The characteristics that obtained substantial values for selection gain (GS%) were those involved with fruit color L (32.52%), °Hue (27.91%), and C (14.68%), together with mass total fruit (TFM = 13.75%), commercial fruit mass (CFM = 18.09%) and commercial fruit number (CFN = 18.42%), when compared to the characteristics average commercial fruit mass (ACFM), total soluble solids (TSS) and firmness (F), which presented low values for this same parameter (3.72%, 0.16% and 1.98%, respectively).

It was possible to select 44 genotypes superior to the 13 control genotypes in the segregated populations, according to the selection index of Mulamba and Mock (1978), with a selection intensity of 3%. According to the analysis of the distribution of the evaluated characteristics data (Figure 1), there was less variability between the genotypes and the control cultivars, for the attributes related to TFM, CFM, and CFN fruits, with a better distribution around the median. For the color attributes of L, °Hue, and C fruits, data dispersion and variability were higher between genotypes and control.

It is possible to observe the presence of outliers at the upper limit in the unselected genotypes for all traits, mainly for CFN, ACFM, and TSS (Figure 1), which is reduced when only selected genotypes and controls are observed. However, for most of them, except for TSS and C characters, the median values found for unselected and selected genotypes were higher than for controls (commercial cultivars plus parents) and selected individuals in all variables.

Of the 44 genotypes, the individual indicated by the methodology with the best values for the evaluated variables was RVDA11CR59. The RVCS44CR population had the most individuals selected (16), corresponding to about 36% of the total number of individuals, followed by RVFS07CR (12), RVDA11CR (6), RVCA16CR (6), and RVFS06CR (4).

### 2.2. Multivariate Analyses among Selected Genotypes

The heatmap containing the cluster analyses of the genotypes and traits (Figure 2) revealed two distinct groups among the individuals indicated by the selection index. The genotypes that obtained high responses for CFN, CFM, TFM, ACFM, and C were grouped. All the populations studied in this group and the genotype ranked highest by selection index are represented, totaling 15 genotypes. In the second group, 29 genotypes were grouped to obtain higher responses for °Hue, L, TSS, and F.

The same pattern can be observed in PCA (Figure 3). The biplot indicated the same groups, showing that the first two principal components (PC) explained 55% of the variation between the selected and control populations (PC1 = 31.9% and PC2 = 23.1%). In the two quadrants with positive values for PC1, cluster 1 obtained high responses for the production traits CFN, CFM, TFM, ACFM, and C, while cluster 2 proved to be more responsive to °Hue, L, TSS, and F arranged near the positive quadrant for PC2. However, the traits L, TSS, F, and ACFM did not contribute sufficiently to explain the variance of the genotypes observed. Also, the population that obtained the highest frequency in cluster 1 (6 genotypes) and cluster 2 (10 genotypes) was RVCS44CR, representing 40% and 35% of the total individuals in this group, respectively.

The correlation network analysis among the variables observed in this study (Figure 4) indicates a strong positive correlation between TFM, CFM, and CFN, with estimated numbers ranging from 0.67 to 0.88. This indicates that these attributes are interconnected and act directly on the productive capacity of the genotypes evaluated. As for the post-harvest variables, °Hue has a median negative correlation with TFM and CFM, in the same way as between C and L, indicating that these traits can be considered inversely proportional. The Pearson’s correlation matrix’s lower and upper limits were −0.59 and +0.88, respectively.

The multivariate analyses performed here, whether cluster analysis (Figure 2), PCA (Figure 3), or CNA (Figure 4), showed that the genotypes in cluster 1 have divergent values compared to others grouped in cluster 2, indicating that the values obtained to TFM, CFM, CFN, and ACFM along with C were preponderant for the grouping of these individuals. Thus, it is possible to observe that the patterns observed for the variables evaluated among the selected genotypes differed, although superior to the controls used in this study.

## 3. Discussion

Strawberry cultivation is widespread worldwide for consumption in natura or in processed form. Breeding programs are underway to develop new materials that meet expectations and improve agronomic characteristics such as yield and fruit quality. The need to increase the indicators of characteristics of economic interest in plants adapted to increasingly extreme agroclimatic zoning stimulates the search for new strawberry genotypes that contribute to the solution of a series of problems related to the production and quality of the fruits, in addition to the adaptability of varieties [19]. An essential demand is to provide more short-day cultivars that provide fruit with higher quality and yield. However, the selection process is complex because, besides being octoploid, most characters have polygenic inheritance [20,21,22]. Another factor that makes breeders’ work hard is grouping agronomic and post-harvest characters in a single cultivar in a balanced way. To achieve this balance, the creator must use more efficient statistical tools.

Analysis of data distribution (Figure 1) indicates that all genotypes had higher and better distributed median values than commercial cultivars and their parents for fruit production traits, while the opposite occurred for all color traits. This shows that the responses of the genotypes regarding fruit color were quite variable. However, the coefficients of genotypic variation indicate that environmental conditions are much less affected. Characteristics related to appearance can be defined as phenotypic classes that are easily distinguishable and little influenced by the environment, as one or a few genes mainly control these characteristics [10,23].

There was high variability and significant difference between most of the evaluated characteristics, except for the number and total fruit mass, for all evaluated genotypes (Table 1). These data indicate significant divergence from other studies that evaluated strawberry genotypes in terms of product characteristics and fruit quality. Some studies found differences for all evaluated properties [8,9,10,11,12,24]. However, others showed that the attributes did not differ when evaluating the aspects involving production, post-harvest, and physical–chemical variables [7,25,26]. This difference can be attributed to the genotypic diversity presented by the species. Commercial cultivars originating from the same crossing indicated low similarity among themselves when compared by morpho–agronomic descriptors [27], demonstrating that using these materials as genitors is possible in a promising way without loss of genetic variability. In addition, the management or system chosen for cultivation can interfere with choosing the best crosses to obtain genotypes and contribute to genetic dissimilarity between populations [28]. Thus, for genetic gain in a breeding program, there must be high genetic variability available among populations of the species. This work indicates that contrasting parents can achieve genotypes with superior attributes.

Furthermore, the heritability was significant when comparing the results with other studies that evaluated strawberry populations for the same purpose. This study observed that all traits related to fruit production and quality have high magnitudes of heritability (Table 1), especially those related to fruit color (L, °Hue, and C). Similar values for this genetic parameter were observed for total soluble solids, color, and firmness when studying day-neutral progenies and short-day strawberry cultivars in Korea [29] and F_1_ genotypes in a breeding program reproduction in Japan [30]. On the other hand, in different regions of Brazil, some studies obtained high heritability values (above 80%) for yield and physicochemical traits [7,24,26]. Also, when the genetics of several traits in progeny between elite *F. virginiana* selections and *F. × ananassa* cultivars and selections were investigated in Michigan, Minnesota, and Ontario, the genotypes with the largest fruit and highest fertility were the most productive, and their photoperiod sensitivity had little bearing on their productivity [31]. This suggests that the genetics of these traits are complex and influenced by multiple genes and environmental factors, including photoperiod. The expression of production and quality traits can be highly influenced by the environmental conditions of cultivation and the location where the experiment is developed, which can be verified in the values found here. Besides the heritability, the coefficients of genotypic variation found were sufficient to justify the selection of strawberry genotypes for the evaluated traits. Also, the CV_g_/CV_e_ ratio values for all traits were above 1, demonstrating that the genetic variation exceeded the environmental variation, another indication of selection feasibility. The chances of genetic gains in clone selection are more significant when the CV_g_ value is higher [32] and when the CV_g_/CV_e_ ratio is close to 1 [33].

With the estimates of selection gains of the observed traits (Table 1), it was possible to notice that although the average commercial mass, total soluble solids, and fruit firmness present high heritability, the selection gain was not expressive. The selection gain was high for the traits of the total mass of fruits, the mass and number of commercial fruits, and the fruit’s color. The heritability of a trait specifies which portion of the total variability is the result of genetic causes and the extent of the rate of genotypic variance in the phenotypic variance. However, the effectiveness and potential of traits under selection can be revealed by evaluating selection gain, where heritability values and genetic advance as a percentage of the mean together are more valuable tools for selection than either alone [34].

Multivariate analyses are essential for selecting parents and cross to obtain superior strawberry genotypes. Through multivariate methods, studies were able to determine the potential of cultivars for greenhouse cultivation [10,15] or even to select genotypes for fresh consumption and processing based on traits of economic interest [7,19]. The selection index of Mulamba and Mock (1978) has been employed for genotype selection in strawberry fruit satisfactorily [7,10,24]. Here in this study, the employment of this selection index with 5% intensity indicated 44 genotypes superior to commercial cultivars among 846 evaluated. The highest-ranked genotype among all selected was ‘RVDA11CR59’, a cross between ‘Camino Real’ × RVDA11 (‘Dover’ × ‘Aromas’). However, the population with the highest frequency was RVCS44CR (16 genotypes). Thus, exploring these individuals’ relationships is necessary since all populations evaluated were represented among the selected ones.

Comparing the analyses (Figure 2, Figure 3 and Figure 4) of the 44 genotypes selected made it possible to form clusters that indicated differences between individuals in distinct groups. Of those selected, 15 genotypes were grouped in cluster 1 (Figure 3) because they shared high responses for production and color intensity traits but low responses for color tone and brightness, total soluble solids, and fruit firmness (Figure 2). The opposite could be observed for the 29 genotypes grouped in cluster 2. Thus, it is possible to observe that even within the selected genotypes, enough difference made it possible to group the selected ones. Moreover, PCA (Figure 3) shows that luminosity, total soluble solids, firmness, and average mass of commercial fruit did not contribute enough as the others to construct PC1 and PC2. This reveals why genotypes with more balanced yield and color intensity attributes were grouped. Similarly, the estimation of the correlation among the variables (Figure 4) indicated a strong positive correlation among the production traits, total fruit mass, commercial fruit mass, and the number of commercial fruits, but this does not apply to the others. 

Hence, the magnitude of the response obtained for the production and post-harvest traits allowed the appropriate grouping, showing that individuals from the population with the highest frequency of selected genotypes RVCS44CR cross between ‘Camino Real’ × RVCS44 (‘Camarosa’ × ‘Sweet Charlie’) have more balanced values among the characters of greater genetic variability. This agrees with previous data that indicated that the cross ‘Camarosa’ × ‘Sweet Charlie’ was the most promising among those evaluated [7,12,25].

The short-day cultivars ‘Camarosa’ and ‘Camino Real’ have high yield averages [5,6], in addition to having high contents of anthocyanin and vitamin C [35,36]. Due to these traits, producers prefer them over day-neutral ones, with greater heat tolerance and longer harvest times [6]. The crosses used here can bring new options of genotypes with balanced responses for yield and fruit quality, although traits related to higher disease resistance [6,37], for example, need to be better investigated. 

High genetic variability among strawberry progenies obtained with intraspecific and interspecific crosses is the primary factor for selecting desirable characters: high yield, fruit size, firmness, and external appearance [38]. With the high variability present in the genotypes evaluated and the low levels of correlation between yield and fruit quality, this study presented the need for the use of methods and evaluations that allow the selection of new materials that are more harmonious with the traits of interest and their respective heritability, as well as good values of selection gain. The selection of genotypes according to the criteria of the breeder or by the genetic variation coefficients attributed to the different weights brought by the Mulamba and Mock index may cause different clusters to be obtained. 

The multivariate analyses helped select and group dissimilarities among strawberry genotypes. It was possible to group the selected genotypes and cover all the populations studied, demonstrating that those with the same group had similar and balanced responses. Consequently, these can be auxiliary tools in deciding which criteria should be adopted to advance genotypes in a strawberry breeding program. Furthermore, the correlation between multiple traits, heritability, and selection gain can guide the breeder in choosing genotypes with more balanced attributes between fruit yield and quality and the most appropriate strawberry breeding methods. 

## 4. Materials and Methods

### 4.1. Experimental Hybrids Obtaining

Crosses were performed according to the methodology proposed by the University of Florida [39]. The genotypes were obtained by crossing the short-day commercial cultivar ‘Camino Real’, used as the female parent, and the first-generation genotypes RVCA16, RVCS44, RVFS06, RVFS07, and RVDA11, obtained from previous selections [7,12], used as the male parents, thus resulting in five segregated populations (Table 2).

After crossing, the achenes were extracted, dried, scarified, and germinated in vitro, as described by the methodology established before [40]. At 40 days after development in the growth chamber, seedlings with four to five true and expanded leaves were transplanted into 50-cell polypropylene trays containing a commercial substrate (CarolinaSoil^®^, Santa Cruz do Sul, RS, Brazil) and kept in a greenhouse at 28 ± 3 °C, 80 ± 5% RH, and a 12-h photophase, for acclimatization.

For the development of the seedlings, three applications of 2 mL per plant were made with the liquid formulation (50% of the distilled medium MS L^−1^ H_2_0). Thirty days before the transplant, the genotypes were submitted to the vernalization process in a cold chamber, with an average temperature of 5 ± 2 °C, RH 80 ± 5%, and photoperiod of 16/8 h.

### 4.2. Experimental Field

The experiment was conducted in autumn/winter in the southern hemisphere at geographic coordinates 24°14′52″ S, 51°41′06″ W, and 1110 m altitude. According to the Köppen classification [41], the climate is humid subtropical Cfb, with hot summers and winters, severe and frequent frosts, and no defined dry season. The average annual temperature is 17 °C, the maximum average annual temperature is 23.5 °C, the minimum annual average temperature is 12.7 °C, and the average annual precipitation is 1946 mm [42].

The seedlings were transplanted into beds 20 cm high and 1.20 m wide, covered with 25 μm thick black polyethylene plastic film (mulching) and 100-micron polyethylene plastic film to form the low tunnels. The spacing used was 0.30 × 0.40 m, forming three rows.

An augmented block experimental design was used [43], with 13 control treatments, of which eight were commercial cultivars (‘Camarosa,’ ‘Camino Real,’ ‘Monterrey,’ ‘Albion,’ ‘Festival,’ ‘Dover,’ ‘Aromas’ and ‘San Andreas’) and five were pre-selected genotypes used as parents (RVFS06, RVCA16, RVCS44, RVFC07, RVDA11). A total of 1500 experimental genotypes was arranged in five blocks, according to segregating populations.

According to soil chemical analysis, the beds were fertilized with 400 kg per hectare of NPK fertilizer (4-14-8). Complementary fertilizations with 20-00-20 fertilizer were carried out every 30 days after pre-flowering (40 to 50 days after transplanting), using 200 kg per hectare, divided into five times. Boric acid and zinc sulfate were applied via foliar application at concentrations of 1% and 2% at the beginning of flowering, respectively. Calcium chloride was sprayed at 0.4% every 15 days during the fruit production phase. The irrigation system used was located with drippers spaced 20 cm apart.

For phytosanitary control, fortnightly preventive sprayings were carried out, interspersed between the products Abamectin (75 mL ha^−1^), Thiametoxan (10 mL ha^−1^), and Fipronil (250 mL ha^−1^). Control of fungal diseases was performed with applications of Azoxystrobin (16 g ha^−1^), Tebuconazole (75 g ha^−1^), and Mancozeb (250 g ha^−1^) alternately.

### 4.3. Production and Post-Harvest Traits Assessment

The harvests were done weekly, with dark red fruits considered ripe. The harvested fruits were evaluated for production characteristics: total fruit mass (TFM); total number of fruits (TFN); average fruit mass (AFM, g/fruit); commercial fruit mass (CFM, g/plant); fruit commercial number (CFN); fruit average commercial mass (ACFM, g/fruit). Fruits larger than 35 mm were considered marketable.

Commercial fruits were sampled at each harvest with the same maturation pattern for post-harvest analyses. Variables related to post-harvest were evaluated. Total soluble solids (TSS) data were obtained by direct reading in a bench refractometer (Optech model RMT, San Prospero, MO, Italy), using homogenized and filtered pulp at room temperature, obtaining the values in degrees Brix. Firmness (F) was measured in three fresh fruits, per harvest, from two points equidistant from the center of the fruit, using a penetrometer with a 3 mm Instrutherm tip (model PTR-300). The average of the collections was calculated, and the results were expressed in Newtons (N). The values of luminosity (L), hue angle (°Hue), and chroma (C), referring to the external coloration of the fruits, were obtained using a Minolta CR-410 colorimeter. The values of L, a (red/green coordinate), and b (yellow/blue coordinate) were obtained at three different points on each fruit. With the values of coordinates *a* and *b*, it was possible to obtain the parameters related to the hue angle (°Hue) and purity or intensity of the color (C) with the equations [44]:(1)$${}^{\circ}Hue={tan}^{-1}\left(\frac{b}{a}\right)$$
and
(2)$$chroma=\sqrt{a^{2}+b^{2}}$$

### 4.4. Statistical Analyses

The analysis of variance and estimation of genetic parameters were performed using the Genes program [45]. In Federer’s augmented block scheme, the control treatments are the ones that allow the estimation of the error and the formation of complete blocks with control treatments and the inclusion in the blocks of genotypes to be tested, with each genotype appearing in a block [46].

The statistical model used in augmented blocks is: (3)Yij=μ+τi+Bj+εij,
where *Y**i**j* is the character value for the i-th treatment in the j-th block; *μ* is the overall mean; *τ**i* is the effect of the i-th treatment (decomposed into Ti: effect of the i-th control with i = 1, 2…13 and Gi: effect of the i-th genotype with i = 1, 2…1500); *B**j* is the effect of the j-th block, with j = 1, 2…5; *ε**i**j* is the random error.

The heritability, phenotypic, genotypic, and environmental coefficient of variation were estimated for applying the classical index [45]. The expected selection gain (GS (%)) for the selected hybrids about the set of hybrids was obtained by the following expression:(4)$$GS\left(\%\right)=\frac{\left[\left(Xs–Xo\right)h^{2}\right]\cdot 100}{Xo},$$
where Xs is the mean of the selected hybrids, Xo is the mean of the hybrids, and h^2^ is the broad heritability, calculated by the expression represented by: (5)$$h^{2}=\frac{V_{g}}{V_{f}},$$

So that V_g_ is the effect of genotypic variance and V_f_ is the effect of phenotypic variance.

For selecting the experimental genotypes [47,48], the non-parametric index based on the sum of ranks [14] was used, with a selection intensity of 5%. This index ranks the genotypes of each trait individually, assigning absolute values according to the classification direction determined by the improver, from highest to lowest or vice versa, according to whether this direction is favorable for improvement. 

The values assigned in the classification of each characteristic are added, resulting in the selection index. The expression obtains this index:(6)I=r1+r2+…+rn,
where *I* is the index value for a given individual, r_j_ is an individual’s classification (rank) about the j-th variable, and n is the number of traits considered in the index. This procedure allows the ranking order of the traits to have different weights, as specified by the breeder. Thus, the expression
(7)I=p1r1+p2r2+…+pnrn,
the p_j_ is the economic weight assigned by the breeder to the j-th trait [20], defined in this work as the values of the coefficients of genetic variation (CV_g_, %), estimated by ANOVA.

The superior genotypes selected by the selection index used were submitted to multivariate analyses for comparison among themselves, using the responses to the traits evaluated. All multivariate analyses were performed using supplementary packages in RStudio software (version 4.0.2)) [49].

The mean data of the observed variables were adjusted using the following expression: (8)$$Z_{i}=\frac{{(x}_{i}-\bar{x})}{s},$$
where x_i_ is the value obtained in the variable, $\bar{x}$ is the sample mean, and s is the sample standard deviation. This measure indicates how much the standard deviation is above or below the mean and standardizes data in different scales. The Z-score ranged from −3 to +3, reflecting a normal standard deviation (variation of the entire standard normal distribution), a normal distribution with a mean equal to zero, and a standard deviation equal to one. 

After the transformation, a heatmap [50,51] was constructed to visualize the data’s patterns and correlations. In the exact figure, the dissimilarity between the genotypes and the dissimilarity between the evaluated traits were presented, measured by the Euclidean distance, and represented in the form of dendrograms built based on a hierarchical UPGMA (unweighted pair–group method using arithmetic averages) cluster analysis. Also, the data were submitted to a principal component analysis (PCA) [52,53] to verify the contribution of variables and genotypes to constructing the principal components. These were submitted to a CNA (correlation-based network analysis) [53,54,55], built based on Pearson’s correlation matrix to investigate the correlation between production and post-harvest traits.

## 5. Conclusions

Although the population RVCS44CR was the best represented among the selected genotypes, the individual with the highest fitness for fresh consumption evaluated in this study was RVDA11CR59, from the crossing ‘Camino Real’ × RVDA11 (‘Dover’ × ‘Aromas’).

The multivariate analyses selected and correlated strawberry genotypes from crosses of commercial cultivars according to production traits and fruit quality. It was also possible to group the selected genotypes and cover all the populations studied, showing that those sharing the same group had similar and balanced responses.

From 1500 genotypes tested, the selection index proposed by Mulamba and Mock (1978) allowed the selection of 44 genotypes with superior traits to the control treatments evaluated, highlighting the population RVCS44CR, resulting from the cross between ‘Camino Real’ × RVCS44 (‘Camarosa’ × ‘Sweet Charlie’). The species’ genetic diversity should still be continuously accessed, along with evaluating these materials under contrasting environmental conditions, with more repetitions, as a more significant number of genotypes among the selected ones to obtain better phenotypical characterization.

## Figures and Tables

**Figure 1 plants-12-02650-f001:**
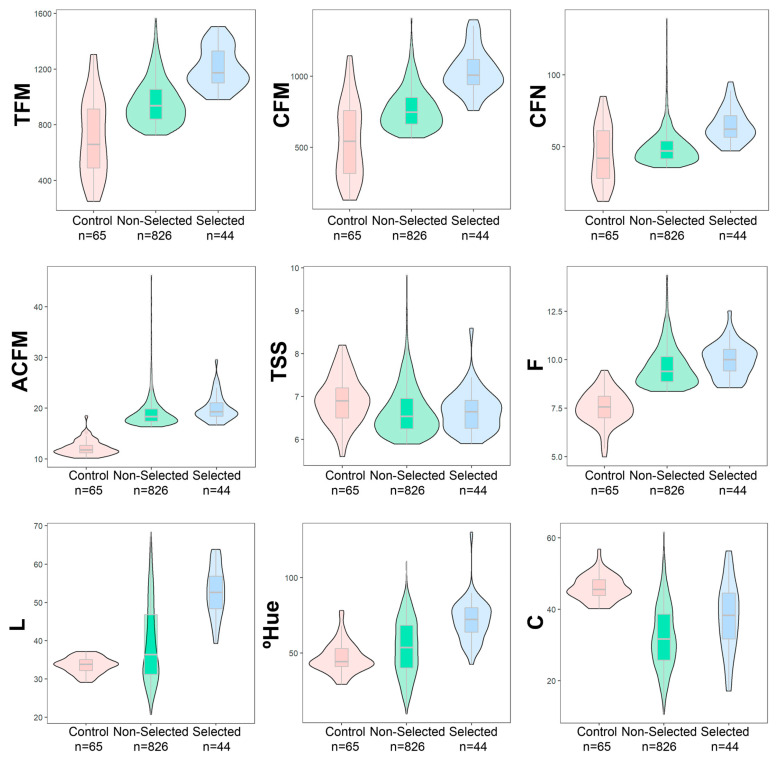
Mean scores boxplots of the selected hybrids and the controls for each trait evaluated. Total fruit mass (TFM); commercial fruit mass (CFM); commercial fruit number (CFN); average commercial fruit mass (ACFM); total soluble solids (TSS); firmness (F); lightness (L); hue angle (°Hue); and chroma (C).

**Figure 2 plants-12-02650-f002:**
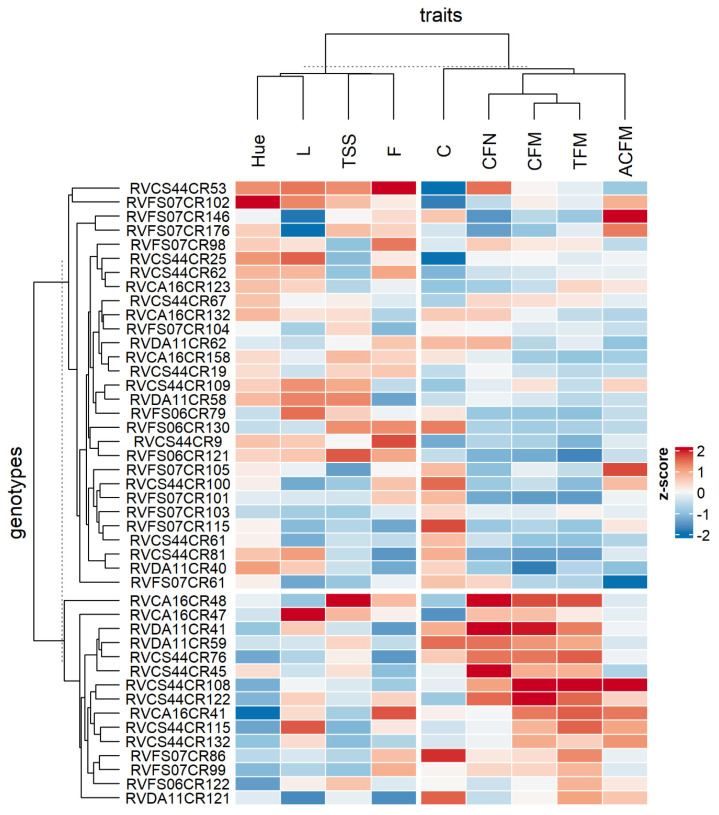
Heatmap presenting the correlation among 47 strawberry populations and traits analyzed. Total fruit mass (TFM); commercial fruit mass (CFM); commercial fruit number (CFN); average commercial fruit mass (ACFM); total soluble solids (TSS); firmness (F); lightness (L); hue angle (°Hue); and chroma (C).

**Figure 3 plants-12-02650-f003:**
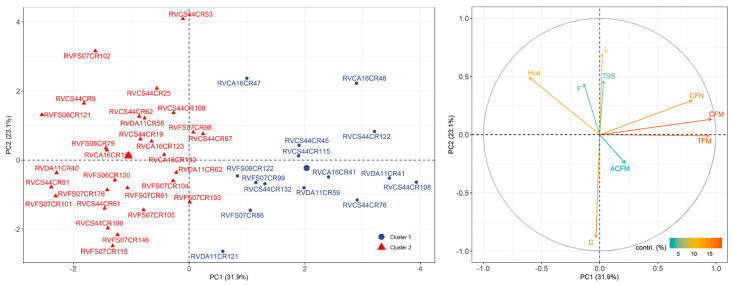
Biplot of the principal component analysis (PCA) among 44 strawberry genotypes and the contributions of the evaluated traits: total fruit mass (TFM); commercial fruit mass (CFM); commercial fruit number (CFN); average commercial fruit mass (ACFM); total soluble solids (TSS); firmness (F); lightness (L); hue angle (°Hue); and chroma (C).

**Figure 4 plants-12-02650-f004:**
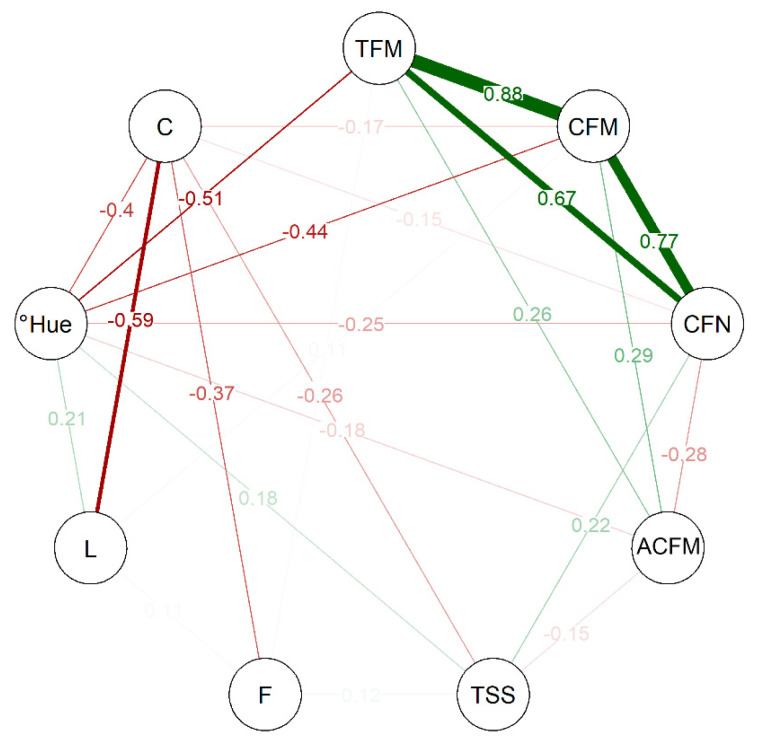
Correlation-based network analysis (CNA) for production and post-harvest traits in strawberries. Total fruit mass (TFM); commercial fruit mass (CFM); commercial fruit number (CFN); average commercial fruit mass (ACFM); total soluble solids (TSS); firmness (F); lightness (L); hue angle (°Hue); and chroma (C).

**Table 1 plants-12-02650-t001:** Estimated genetic parameters for nine agronomic traits in strawberry hybrids observed by the selection index.

Genetic Parameters	TFM **	AFM **	CFN **	ACFM **	TSS **	F **	L **	°Hue **	C **
µ *	975.89	69.8	14.49	769.8	49.89	18.64	6.68	0.51	39.48
µF	972.72	69.58	14.78	781.73	50.1	19.02	6.66	9.62	39.85
CV_controls/genotypes_ (%)	11.90	18.96	10.89	13.78	16.12	7.39	5.69	5.95	2.28
CV_controls_ (%)	16.09	17.85	16.67	19.33	18.03	11.31	5.49	7.51	2.68
CV_genotypes_ (%)	11.67	19.05	10.62	13.49	15.99	7.21	5.71	5.86	2.26
$\sigma_{p}^{2}$	29,096.89	24,564.06	160.38	7.63	0.31	0.98	114.72	344.01	80.25
$\sigma_{e}^{2}$	12,997.73	11,262.51	64.69	1.9	0.14	0.32	0.81	30.83	5.05
$\sigma_{g}^{2}$	16,098.97	13,301.55	95.69	5.73	0.17	0.66	113.91	313.17	75.7
$h^{2}$ (%)	55.32	54.15	59.66	75.09	54.02	67.49	99.29	91.03	93.7
CVg (%)	12.99	14.68	19.45	12.51	6.19	8.45	26.73	32.31	26.64
CVg/CVe	1.11	1.08	1.21	1.73	1.08	1.44	11.84	3.19	3.86
Xo	976.67	785.21	50.19	19.12	6.66	9.66	39.91	54.84	32.55
Xs	1219.45	1047.56	65.68	20.07	6.68	9.94	52.98	71.66	37.65
GG	134.33	142.06	9.24	0.71	0.01	0.19	12.98	15.31	4.78
GS%	13.75	18.09	18.42	3.72	0.16	1.98	32.52	27.91	14.68

* µ: grand mean; µF: weighted average (Federer); CV: coefficient of variation; $\sigma_{p}^{2}$: phenotypical variance; $\sigma_{e}^{2}$: environmental variance; $\sigma_{g}^{2}$: genotypical variance; total fruit mass (TFM); average fruit mass (AFM); commercial fruit number (CFN); average commercial fruit mass (ACFM); total soluble solids (TSS); firmness (F); lightness (L); hue angle (°Hue); chroma (C); Xo: average of controls; Xs: average of hybrids; GG: genetic gain; GS% = genetic gain in percentage; ** significant at 1% by the F-test.

**Table 2 plants-12-02650-t002:** List the male and female genitors used in the hybridization process and their respective numbers of seedlings.

Population	Female Parent	Male Parent	Individuals
RVFS07CR	‘Camino Real’	RVFS07 (‘Festival’ × ‘Sweet Charlie’)	194
RVFS06CR	‘Camino Real’	RVFS06 (‘Festival’ × ‘Sweet Charlie’)	171
RVCA16CR	‘Camino Real’	RVCA16 (‘Camarosa’ × ‘Aromas’)	163
RVCS44CR	‘Camino Real’	RVCS44 (‘Camarosa’ × ‘Sweet Charlie’)	152
RVDA11CR	‘Camino Real’	RVDA11 (‘Dover’ × ‘Aromas’)	190

## Data Availability

All datasets presented in this study are included in the article and can be provided by the authors upon reasonable request.

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
