# Peer review of "Selection of Short-Day Strawberry Genotypes through Multivariate Analysis"

_plants, 2023, doi:10.3390/plants12142650_

Round 1

Reviewer 1 Report

It's good manucript 

Author Response

Response to Reviwer 1

  • It's good manucript 

Answer: Thank you for all your suggestions and corrections.

All the changes were made and highlighted in the last version submitted.

Reviewer 2 Report

The manuscript is well written and the research is well established with correct materials and methods.

Just two comments that are important for the overall evaluation of the submitted research.

1 – In any part of the manuscript we were able to have biometric data of the different genotypes. The use of the statistical tools is well applied but in order to do a correct evaluation we should have the biometric data for all the genotypes under evaluation. This is the only way to compare with past or future research.

2 – Another important issue for a complete comparison is that authors should have done a genome and phenotypic selection. This was not the aim of the research but it would improve it tremendously.

Author Response

Response to Reviewer 2

The manuscript is well written and the research is well established with correct materials and methods.

Answer: Thank you.

Just two comments that are important for the overall evaluation of the submitted research.

1 – In any part of the manuscript we were able to have biometric data of the different genotypes. The use of the statistical tools is well applied but in order to do a correct evaluation we should have the biometric data for all the genotypes under evaluation. This is the only way to compare with past or future research.

Answer: Due to the high number of evaluated genotypes, 1500 experimental hybrids plus the controls, the work aimed to estimate the genetic parameters of the nine traits and use multivariate analysis to promote a better presentation of the data. However, according to the conclusions of the work, from the 40 selected genotypes, new experiments will be carried out in different environments, and various biometric and molecular data will be collected for comparison.

2 – Another important issue for a complete comparison is that authors should have done a genome and phenotypic selection. This was not the aim of the research but it would improve it tremendously.

Answer: We agree with the reviewer that expanding the analyzed variables would give more consistency to the results. However, due to the high number of genotypes evaluated, 1500 experimental hybrids plus controls, it became impossible to evaluate biometric data in this population. In addition, the objective of the work was to estimate the genetic parameters of the nine traits and use multivariate analysis to promote a better presentation of the data and ensure a more balanced selection of genotypes. However, according to the conclusions of the work, of the 40 selected genotypes, new experiments are being carried out in different environments, and diverse biometric and molecular data are being collected for comparison.

Reviewer 3 Report

Very well-written. I was hesitant to review this manuscript since I do not have a background in genetics, but after reading the abstract it seemed that it would be very readable for someone with a solid plant background like myself.

I found only a few small errors or suggestions.

starting with Line 45 - be consistent with the term day-neutral, which you sometimes refer to as neutral-day

Line 106 32,31 should be 32.31

Table 1 caption: GG% = Genetic Gain in Percentage - should be GS%

Line 271 The comparing analyses should be Comparing the analyses

Line 292-294. I found this sentence confusing. I don't think you mean "Despite these traits"

Line 307 change singular dissimilarity to plural (suggestion)

Line 310 change singular criterion to plural criteria

I really liked your data presentation in box plots and heat maps.

Author Response

Response to Reviewer 3

Very well-written. I was hesitant to review this manuscript since I do not have a background in genetics, but after reading the abstract it seemed that it would be very readable for someone with a solid plant background like myself.

Answer: We appreciated your feedback.

I found only a few small errors or suggestions.

starting with

Line 45 - be consistent with the term day-neutral, which you sometimes refer to as neutral-day

Answer: We changed all the mentions of this term in the manuscript.

Line 106 32,31 should be 32.31

Answer: We corrected this number.

Table 1 caption: GG% = Genetic Gain in Percentage - should be GS%

Answer: We also included some corrections in this table highlighted in the last manuscript, including the abovementioned correction.

Line 271 The comparing analyses should be Comparing the analyses

Answer: We changed this sentence.

Line 292-294. I found this sentence confusing. I don't think you mean "Despite these traits"

Answer: We changed to this sentence: “The short-day cultivars 'Camarosa' and 'Camino Real' have high yield averages [5,6], besides having high contents of anthocyanin and vitamin C [35,36]. Due to these traits, producers prefer them over day-neutral ones, with greater heat tolerance and longer harvest times [6].” We also agree that this version makes more sense.  

Line 307 change singular dissimilarity to plural (suggestion)

Answer: We changed to the plural version.

Line 310 change singular criterion to plural criteria

Answer: We changed to the plural version.

I really liked your data presentation in box plots and heat maps.

Answer: Thank you for the observation. Those graphs help a lot when we have an extensive data set like the one presented in this study. Thank you for all your suggestions and corrections. All the changes were made and highlighted in the last version submitted.